# Activity of Experimental Mouthwashes and Gels Containing DNA-RNA and Bioactive Molecules against the Oxidative Stress of Oral Soft Tissues: The Importance of Formulations. A Bioreactor-Based Reconstituted Human Oral Epithelium Model

**DOI:** 10.3390/molecules26102976

**Published:** 2021-05-17

**Authors:** Andrei C. Ionescu, Elena Vezzoli, Vincenzo Conte, Patrizia Sartori, Patrizia Procacci, Eugenio Brambilla

**Affiliations:** 1Oral Microbiology and Biomaterials Laboratory, Department of Biomedical, Surgical and Dental Sciences, Università degli Studi di Milano, via Pascal, 36, 20133 Milan, Italy; eugenio.brambilla@unimi.it; 2Electron Microscopy Laboratory, Department of Biomedical Sciences for Health, Università degli Studi di Milano, via G. Colombo, 71, 20133 Milan, Italy; elena.vezzoli@unimi.it (E.V.); vincenzo.conte@unimi.it (V.C.); patrizia.sartori@unimi.it (P.S.); patrizia.procacci@unimi.it (P.P.)

**Keywords:** DNA, RNA, bioactive compounds, excipients, surfactants, mouthwash, oral gels, oxidative stress, reconstituted oral epithelium, bioreactors

## Abstract

Background: DNA-RNA compounds have shown promising protection against cell oxidative stress. This study aimed to assess the cytotoxicity, protective, or preventive effect of different experimental formulations on oral epithelia’s oxidative stress in vitro. Methods: Reconstituted human oral epithelia (RHOE) were grown air-lifted in a continuous-flow bioreactor. Mouthwashes and gels containing DNA-RNA compounds and other bioactive molecules were tested on a model of oxidative stress generated by hydrogen peroxide treatment. Epithelia viability was evaluated using a biochemical MTT-based assay and confocal microscopy; structural and ultrastructural morphology was evaluated by light microscopy and TEM. Results: DNA-RNA showed non-cytotoxic activity and effectively protected against oxidative stress, but did not help in its prevention. Gel formulations did not express adequate activity compared to the mouthwashes. Excipients played a fundamental role in enhancing or even decreasing the bioactive molecules’ effect. Conclusion: A mouthwash formulation with hydrolyzed DNA-RNA effectively protected against oxidative stress without additional enhancement by other bioactive molecules. Active compounds, such as hyaluronic acid, β-Glucan, allantoin, bisabolol, ruscogenin, and essential oils, showed a protective effect against oxidative stress, which was not synergistic with the one of DNA-RNA. Incorporation of surfactant agents showed a reduced, yet significant, cytotoxic effect.

## 1. Introduction

Reactive oxygen species (ROS) include oxidants of natural origin involved as signaling molecules in many cellular pathways that, under normal circumstances, are essential to life. An imbalance between ROS production and the antioxidant defenses that protect cells is referred to as oxidative stress. This imbalance initiates the disruption of cellular redox signaling and control, and leads to molecular damage [1]. An overproduction of ROS or, especially, their exogenous presence can indeed cause harmful oxidative stress that can disrupt normal physiology [2]. ROS have been shown to damage biomolecules essential to cell functioning, such as DNA, proteins, and membrane lipids. In particular, ROS activity yielding DNA damage can arrest cell proliferation and differentiation, and can trigger apoptosis or directly cause cell death. ROS have been shown to contribute to cancer development by promoting potentially mutagenic DNA changes and damaging lipids in cell membranes [3,4,5,6,7].

Membranes are, in fact, one of the preferential targets of ROS, causing lipid peroxidation. This process alters membrane properties, such as its fluidity, which is fundamental to modulate membrane protein localization and function [8]. Oxidative stress was also reported to increase RNA damage, especially in patients suffering from degenerative diseases [9]. Studies on RNA oxidation found that, under similar conditions, the oxidative damage levels to RNA are usually higher than those to DNA, impairing protein synthesis and all other RNA functions [9]. Counteracting the multitude of different types of ROS-induced DNA and RNA damage is, consequently, a significant challenge for organisms, and a cell without any repair capacity would hardly remain viable.

Oral tissues are exposed to a vast amount of substances, many of which have been shown to produce significant levels of oxidative stress daily. Among these substances, the primary exogenous sources of oxidative stress in the oral cavity are tobacco, alcohol, drugs (e.g., cyclosporine, tacrolimus, gentamycin), cooking (e.g., smoked meat, repeatedly-heated cooking oil), fat, and radiation [10,11].

Therefore, the supply of an exogenous antioxidant may seem beneficial to oral tissues when the detoxifying and repair mechanisms are insufficient to counteract the regular exposure to oxidative stress or when exogenous factors shift the balance towards ROS presence.

One of the possibilities in this sense is represented by formulations containing bioactive molecules, such as hydrolyzed DNA and RNA. Such molecules can pass through the cell membrane by pinocytosis and act as a donor of purine and pyrimidine bases [12]. There is evidence about their ability to increase cell proliferation and activity by working in synergy with several growth factors and influencing immunological response [13,14,15,16]. The result is a repairing activity in cells under oxidative stress; however, the evidence about their effect on mucosa and oral epithelium is relatively scarce. A recent in vitro study on reconstituted human oral epithelium (RHOE) demonstrated that exposure to sodium-DNA-containing solutions showed protective activity against the cytotoxic effect expressed by chlorhexidine gluconate. A side result of the study was that sodium-DNA could protect against cellular damage caused by oxidative stress [17].

This study, therefore, aimed to assess the cytotoxic effect of different experimental formulations (mouthwashes and oral gels) containing hydrolyzed DNA and RNA, and their protective, or preventive activity against oxidative stress in oral epithelia in vitro. The null hypothesis was that the tested formulations would not have any effect on the epithelia.

## 2. Results

The present study tested the effect of several formulations, depicted in Table 1. Formulations A to C represent oral mouthwash, while D and E are gels for topical oral mucosa use. B and E are the placebo counterparts of the mouthwash A and gel D, respectively. At the same time, formulation C represents a “complete” mouthwash formulation containing several other active principles commonly found in mouthwashes with anti-inflammatory and repairing purposes for oral mucosa. An additional solution (phosphate-buffered saline, PBS) was used as a reference (baseline) in all tests.

A bioreactor (Figure 1) was used to grow 3D reconstituted human oral epithelia (RHOE) on coupons lifted at the air/liquid interface. Coupons were then exposed to three different treatments to evaluate the cytotoxicity (Treatment 1), protective effect (Treatment 2), and preventive effect (Treatment 3) of the tested formulations (Figure 2).

The effects were evaluated in terms of cell viability and metabolic activity (MTT assay and confocal laser-scanning microscopy, CLSM) and morphological alterations. The latter was evaluated at structural and ultrastructural levels using light microscopy (630x) and TEM (1600x), respectively.

### 2.1. MTT Assay

The results of the MTT assay are displayed in Figure 3. The first treatment, evaluating possible cytotoxic effects, showed that C and D formulations were not significantly different from the baseline (PBS). A slightly but significantly lower viability was found after treatment with A, B, and E, suggesting cytotoxic effects. 

The possible protective/repairing effects of the formulations against oxidative stress were evaluated in the second treatment group. B and E showed a high reduction in viability that was comparable to the baseline (PBS). This result indicated the absence of a protective activity by such formulations against oxidative stress. On the contrary, all formulations containing hydrolyzed DNA-RNA (A, C, and D) maintained high viability values. 

The third treatment evaluated a possible preventive effect of the tested formulations in reducing oxidative stress damage by exposing the epithelia to the formulations before simulating oxidative stress. All specimens, including PBS reference, showed a significant and equal reduction in viability, meaning that the tested formulations showed no preventive effect on RHOE.

### 2.2. CLSM Observations

Confocal microscopy reconstructions are shown in Figure 4. Cell membranes were stained in green by Syto9, while their nuclei were stained by DAPI and shown in blue. Each row belongs to a different tested formulation, while each column represents one of the three treatments performed. Treatment 1 specimens showed that the dye penetration was limited to the first and most external layer, due to the tight junctions between the cells of the inner layers. All epithelia showed a generally regular aspect with high viability, similar to the baseline (PBS). Formulations A, B, and E showed a slightly higher presence of nuclei on the epithelium surface, indicating initial cell suffering in the most external layer. Treatment 2 specimens treated with B and E showed many dead cells in the external cell layers. In particular, formulation B showed a higher amount of dead cells compared to the PBS reference. This finding is a consequence of the oxidative stress to which the RHOE specimens were subjected. Surprisingly, specimens exposed to A and C formulations following oxidative stress showed regular aspects with high viability, similar to specimens belonging to Treatment 1. These formulations demonstrated a protective activity against oxidative stress. Treatment 3 specimens, including the PBS reference, generally showed low viability, with signs of enlargement of intercellular spaces and disgregation of the external epithelium layers. Formulation A showed slightly higher viability than the other groups. The results of this treatment group demonstrate that there was no preventive effect of the tested formulations against oxidative stress.

### 2.3. Histological Evaluation

RHOE tissue semi-thin sections obtained for each group are shown in Figure 5. Regarding the first treatment, the baseline (PBS) showed complete preservation of tissue structures. Formulations A, B, and C showed very slight alterations of the epithelium structures, such as initial vacuolization. Formulations D and E showed marked vacuolization and enlargement of intercellular spaces, suggesting a cytotoxic effect. Treatment 2 specimens displayed, as expected, extensive damage to RHOE cells, such as marked vacuolization, degenerated nuclei, and enlargement of the intercellular spaces, due to the exposure to hydrogen peroxide. These alterations were most noticeable in specimens treated with the gel formulations (D and E). The mouthwash formulations showed less intense cell structure alterations than the gels, mainly showing vacuolization while the overall epithelium structure was maintained. In particular, formulation A showed signs of cellular suffering limited to the outermost and basal cell layers. Cells not showing degeneration signs were found in the innermost layers, where they were more protected from ROS generated by hydrogen peroxide. These findings demonstrate protective activity against oxidative stress that was shown by formulation A and, to a lesser extent, B and C. Treatment 3 specimens all displayed massive damage to the epithelia, such as marked vacuolization, degenerated nuclei, and enlargement of the intercellular spaces, due to the exposure to hydrogen peroxide. Specimens treated with formulations A and B showed slightly lower tissue damage compared to the other formulations.

### 2.4. TEM Analysis

Specimens were observed using TEM to investigate the ultrastructure of the RHOE after the different treatments. The effects of the different treatments are displayed in Figure 6, compared to the baseline (PBS). Findings are similar to the histological observations using light microscopy. Signs of vacuolization indicating slight cytotoxicity were found when formulations B and C were applied to the untreated epithelium. Surprisingly, gel formulations (D and E) showed extensive cellular damage after Treatment 1. Specimens treated with A, B, and C formulation showed good preservation of tissue structures when the formulation was applied after oxidative stress (Treatment 2). Extensive degradation of epithelium structures (marked vacuolization, degenerated nucleus, and enlargement of the intercellular spaces) was found in treatment 3. In fact, D and E formulations showed signs of degradation of the epithelium structures in all treatments. The baseline (PBS) showed typical epithelium structures with no signs of cellular suffering.

## 3. Discussion

When the oral environment is exposed to oxidative stress, or when exogenous factors shift the balance towards ROS production, the physiological detoxifying and repair mechanisms can be insufficient to counteract the effects of such regular exposure. Therefore, the supply of an exogenous antioxidant may seem beneficial to oral tissues. The results described in the present study demonstrated, for the first time in vitro, that a mouthwash formulation containing hydrolyzed DNA and RNA effectively protected against oxidative stress. The same active principles, when applied before oxidative stress simulation, were not helpful in its prevention.. The tested formulations showed a low, albeit significant, level of cytotoxixity independent from the presence of hydrolyzed DNA and RNA. This effect was due to the presence of surfactants. 

DNA and RNA-derived bioactive molecules are functional compounds recently introduced in medical care. Such compounds are able to increase proliferation and activity of different cell types by acting in synergy with several growth factors (i.e., epidermal growth factor, EGF, platelet-derived growth factor, PdGF, and fibroblast growth factor, FGF), and influencing immunological response [13,14,15,16]. In vitro studies also showed that such molecules stimulate fibroblasts proliferation, promoting collagen, proteoglycans, and elastin synthesis. Other studies suggested a photo-protective efficacy of this molecule, which preserved fibroblasts in cell culture by UV-induced damages through ROS generation. The result of such activity is a repairing action related to the upregulation of cytokines and growth factors in cells under stress or metabolic alteration [15,16,18,19]. The compounds have very low toxicity, high tolerability, and safety, usually incorporated in topical formulations at low concentrations. Our results are consistent with other observations that found a protective effect of a sodium-DNA-containing mouthwash formulation in protecting oral epithelium cells against oxidative stress in a bioreactor-based in vitro model [17].

Surprisingly, the positive effect of hydrolyzed DNA-RNA was not synergistic in combination with other tested bioactive molecules, such as hyaluronic acid, β-Glucan, allantoin, bisabolol, ruscogenin, and several essential oils (Table 2). The aforementioned molecules were proven to have anti-inflammatory, antioxidant, and anti-aging activity, helping in wound healing, epithelial regeneration, and extracellular matrix regeneration [20,21,22,23,24,25,26,27,28,29,30,31,32,33,34]. However, a formulation containing these bioactive molecules in addition to hydrolyzed DNA and RNA did not yield a higher protective effect than the formulation containing DNA and RNA alone. This datum suggests that the tested DNA and RNA compounds already exerted the maximum protective efficacy against oxidative stress under the present experimental conditions.

Dental care products use a variety of bioactive molecules to accomplish the multiple claims that are most often requested. The development process has to optimize formulations often containing active principles with widely different scopes that sometimes clash with each other, ranging from the expected activity to just improvements in acceptability and the need for distinctive flavoring, especially for branding reasons. It is believed that improved clinical efficacy and tolerability of formulations are just as significant as conditioning signals in encouraging patient compliance with the needed treatment [35].

Thus, it is unsurprising that oral formulations are complex mixtures involving a number of excipients in addition to the bioactive molecules. Excipients include humectant in mouthwashes, thickeners and film-formers in gels, preservatives both to maintain bioactive principles and to prevent microbial/fungal proliferation, sweeteners, and flavors (Table 2).

Finally, surfactants are added (Table 2), supporting foaming and an even mixing/distribution of the oral formulations. However, their use brings concerns related to safety issues. For instance, surfactants commonly present in oral formulations, such as Cetearyl Alcohol, Cetyl Palmitate, and Ceteareth-12/20 have been found to produce dermal irritation [36,37,38]. These surfactants incorporated in the tested mouthwash formulations are the only excipients that can be associated with increased cytotoxicity (Table 2) after 30 min exposure, especially seen in formulations A and B when compared to the baseline PBS. These considerations suggest that these excipients may not be indicated in an oral formulation designed for protection against oxidative stress.

Furthermore, a polyethylene glycol derivative of hydrogenated castor oil (PEG-40 Hydrogenated Castor Oil) is a surfactant that can change the barrier properties of human oral mucosa in vitro and in vivo by altering the tight junctions of epithelial cells [39]. This surfactant, added to the tested gel formulations of the present study, was shown to decrease the viability of a cell line of fibroblasts derived from mice (L929) and human cervical cancer cells (HeLa) [40]. It is likely that its effect on cell junctions, especially visible using TEM, may have improved the hydrogen peroxide penetration, thus worsening the effect of ROS on the tested epithelia.

Gel formulations for the oral environment are specifically designed to ensure adequate adherence to tissues, preventing a too easy clearance of the formulation by salivary wash-out or mechanical forces. In this sense, ingredients, such as thickeners and film-formers (Table 2), are routinely used. Our histological data suggested that gel formulation D, containing the same bioactive molecules, did not express similar activity compared to the mouthwash in protecting against oxidative stress. At the same time, MTT results for gel formulation D did not show cytotoxic activity against RHOE. Thickeners and film-formers may have reduced the access and availability of nutrients and oxygen rather than allowing better contact and delivery of the bioactive molecules carried by the gels. This mechanism may explain morphological damages found on RHOE specimens assessed for cytotoxicity after both gel treatments. On the contrary, MTT results showed cytotoxic effects of A, B, and E formulations. In fact, MTT is a colorimetric test based on the reduction of the tetrazolium salt at the plasma membrane level and, to a lesser extent, by cellular redox systems. Therefore, one may speculate that its response may not be initially related to the ultrastructural changes highlighted using TEM analysis, especially after the relatively short exposure time to the different formulations. For the same reason, a relatively high MTT reduction indicating the existence of a protective action against oxidative stress by the test gel formulation was not confirmed by the other morphological observations (CLSM, histological evaluation, and TEM). The use of both biochemical and morphological analytical methods is therefore paramount for an in-depth understanding of the effects exerted by bioactive principles.

The reconstituted human oral epithelium is a practical and recent tool to study the effect of different active compounds on soft tissues, mimicking the behavior of human mucosa and allowing for testing of active compounds in a standardized way. Stratified, cultured TR146 cell layers derived from a human neck metastasis originating from a buccal carcinoma were used to create an in vitro model of RHOE. The resulting stratified epithelium closely resembles normal human buccal epithelium [41]. The bioreactor setup helped reconstitute the oral epithelial tissues in a controlled environment as close as possible to the natural one. A chemically defined medium induced the expression of all physiological markers, thus behaving as in vivo human epithelial cells when treated with pharmacologically active or cytotoxic compounds. The model also exhibits tissue repair mechanisms that reflect the in vivo wound healing processes [41,42,43]. For these reasons, in vitro testing using RHOE and bioreactors helps avoid animal testing and can be used before human testing, allowing to accurately test the activity of newly developed active principles and avoid ethical concerns. However, the model’s limitation can be seen in the present study when using gel formulations. When applied to the RHOE, the latter, contrarily to in vivo situations, reduced the availability of nutrients and oxygen, bringing cell suffering even after applying the placebo gel, as shown in our results. Caution should be used when applying RHOE models to the study of gel formulations since the results thus obtained may misrepresent the clinical setting. Furthermore, measurements of cellular anti-oxidative response elements (for instance, heme oxygenase and superoxide dismutase activities, and GSH levels) may provide further insights on the physiological reactions of oral epithelia to oxidative stress when testing protective formulations.

Nevertheless, and within its limitations, the present study opens the possibility for standardized pre-clinical analysis of the efficacy of bioactive formulations designed for the oral environment.

In conclusion, our results showed that hydrolyzed DNA and RNA solutions effectively protected against oxidative stress but were not helpful in its prevention. Gel formulation containing the same bioactive molecules did not express similar activity compared to the mouthwash solutions; therefore, the null hypothesis was rejected. Most interestingly, excipients played a fundamental role in enhancing or even decreasing the bioactive molecules’ effect. In particular, bioactive molecules, such as hyaluronic acid, β-Glucan, allantoin, bisabolol, ruscogenin, and several essential oils, showed a protective effect against oxidative stress. However, this effect was not synergistic and did not reach the extent of hydrolyzed DNA and RNA. The presence of surfactants seemed the main cause of the cytotoxic effects expressed by the tested formulations, possibly worsening oxidative stress. Therefore, a mouthwash formulation containing hydrolyzed DNA-RNA protected against oxidative stress without additional enhancement by other bioactive molecules.

**Table 2 molecules-26-02976-t002:** Analysis of the compounds used in the tested formulations, together with their documented positive, negative, or neutral effects.

Categories	Compounds	Description	Supposed Positive Effects	Supposed Negative Effects	A	B	C	D	E
**Active Principles**	Hydrolysed DNA RNA	Contains microbial-hydrolyzed low molecular weight fragments of ribonucleic acid (RNA) and deoxyribonucleic acid (DNA) of vegetal origin.	Anti-inflammatory and protective against ROS [17].		✓		✓	✓	
	Hyaluronic acid	Anionic, non-sulfated glycosaminoglycan.	Extracellular matrix regeneration, epithelial regeneration, wound healing [20].				✓	✓	
	Beta-Glucan	High-molecular weight β-D-glucose polysaccharide. Water-soluble natural extract.	Antioxidants and anti-aging activity [21].				✓	✓	
	Allantoin	2,5-Dioxo-4-imidazolidinyl urea. Synthetically produced from uric acid.	Promotes cell proliferation and facilitates wound healing [22,23].				✓	✓	
	Bisabolol	Monocyclic sesquiterpene alcohol. First isolated from Matricaria chamomilla (Asteraceae).	Anti-irritant, anti-inflammatory, and antimicrobial activity [24,25,26,27,28].				✓	✓	
	Ruscogenin	First isolated from *Ruscus aculeatus*, also a major steroidal sapogenin of the traditional Chinese herb Radix Ophiopogon japonicus.	Anti-inflammatory and protective against ROS [29,30].				✓	✓	
**Essential Oils**	Glycyrrhetinic Acid	Oleanoic acid derived from shredded Glychirriza (licorice) roots.	Antioxidant, anti-inflammatory activities [31].				✓	✓	
	Leptospermum Scoparium Branch/Leaf Oil	Essential oil coming from the Manuka tree native to New Zealand.	Antibacterial and antifungal activity [32,33,34].				✓	✓	
	Melaleuca Alternifolia Leaf Oil	Tea tree oil, essential oil distilled from the leaves of a native Australian plant.	Antibacterial, Antimicrobial and Antiviral activity [32].				✓	✓	
**Surfactant Agent**	Ceteareth-12/20	Polyethylene glycol (PEG) ethers of Cetearyl Alcohol		Dermal irritation [38].	✓	✓	✓		
	Cetyl Palmitate	Ester derived from hexadecanoic acid and hexadecanol		Dermal toxicity Dermal irritation Dermal sensitization [37].	✓	✓	✓		
	Cetearyl Alcohol	Straight-chain alcohol. Mixture of mostly Cetyl and Stearyl Alcohols, which are fatty alcohols that occur naturally in small quantities in plants and animals.		Dermal irritation Cytotoxicity [36].	✓	✓	✓		
	PEG-40 Hydrogenated Castor Oil	Polyethylene glycol derivative of hydrogenated castor oil		Alterations of the plasma membranes of epithelial cells Tight junction opening Cytotoxicity [39,40].				✓	✓
	Caprylyl Glycol	1,2-glycol compound with 8 carbons in the carbon chain. Also used as humectant and preservative agent.	Safe for use, no negative effect reported [44].				✓	✓	
	1,2-Hexanediol	1,2-glycol compound with 6 carbons in the carbon chain. Also used as humectant and emollient and preservative agent.	Safe for use, no negative effect reported [44].				✓	✓	
**Emulsifier/Emollient**	Dicaprylyl Ether	Derived from the dehydration of octane. Used as skin conditioner, emollient and solvent.	Safe for use, no negative effect reported [45].		✓	✓	✓		
	Coco-caprylate/caprate	Made by combining esters from coconut-derived fatty alcohol with caprylic and capric acids, also from coconut. Emollient.	Safe for use, no negative effect reported [46].		✓	✓	✓		
	Glyceryl Stearate	Ester of stearic acid and ethylene glycol. Monoglyceride commonly used as an emulsifier in foods.	Safe for use, no negative effect reported [47].		✓	✓	✓		
**Humectant**	Propylene Glycol	Propanediol: propane where the hydrogens at positions 1 and 2 are substituted by hydroxyl groups. Used as an organic solvent and diluent, and to absorb extra water and maintain moisture	Safe for use, no negative effect reported [48].		✓		✓	✓	✓
	Glycerin	Simple polyol compound with three alcohol hydroxyl groups.	Safe for use, no negative effect reported [49].				✓	✓	
**Thickener**	Cellulose Gum	Carboxymethyl cellulose	Non cytotoxic [50].					✓	✓
	Carbomer	Poly-acrylic acid	Non cytotoxic, improved wound healing [51].					✓	✓
**Film-Former**	Calcium/Sodium PVM/MA Copolymer	PVM/MA Copolymer is a copolymer of methyl vinyl ether and maleic anhydride or maleic acid. Used as binder and film-former.	Safe for use, no negative effect reported [52].					✓	
	VP/VA Copolymer	Large molecule made from vinyl pyrrolidone (VP) and vinyl acetate (VA) monomers.	Safe for use, no negative effect reported [53].					✓	
**Sweetener**	Xylitol	Polyol, artificial sweetener	Safe for use, no negative effect reported [54].		✓	✓	✓		
	Sodium saccharin	Artificial sweetener	Safe for use, no negative effect reported [55].		✓	✓	✓	✓	✓
	Ammonium Glycyrrhizate	Natural extract from Glychirrizia plant	Antiviral, anti-inflammatory [56].	Gap-junction inhibitor, cytotoxic [56].				✓	
**Preservative**	o-Cymen-5-ol	Substitute of parabens	Safe for use, no negative effect reported at the tested concentration (0.1%) [57].				✓	✓	
	Sodium Benzoate	Sodium benzoate is the sodium salt of benzoic acid. It is an aromatic compound with antimicrobial activity, therefore is used as a preservative in food products.	Safe for use, no negative effect reported [58].		✓	✓	✓	✓	✓
	Phenoxyethanol	Ether alcohol, aromatic compound with antimicrobial activity. Extensively used as preservative in pharmaceuticals, cosmetics and lubricants.	Safe for use, no negative effect reported [59].		✓	✓	✓	✓	✓
	Citric acid	Tricarboxylic acid found in citrus fruits, Used as a preservative due to its antioxidant properties.	Safe for use, no negative effect reported [60].		✓	✓	✓		
**Cosmetic Colorant**	CI 16255	Ponceau 4R, synthetic colourant used for food colouring	Safe for use, no negative effect reported [61].		✓	✓	✓		
	CI 42090	Brilliant Blue FCF (Blue 1) is a synthetic organic compound used as a colorant for cosmetics and food.	Safe for use, no negative effect reported [61].					✓	✓

## 4. Materials and Methods

### 4.1. Reagents

Reagents, culture media, and disposables used in this study were obtained from Merck (E.Merck AG, Darmstadt, Germany). RHOE specimens (0.5 cm^2^, SkinEthic HOE™/Human Oral Epithelium) were obtained from EPISKIN (EPISKIN, Lyon Cedex 7, France). Test formulations were obtained from Betafarma S.P.A. (Cesano Boscone, Milan, Italy). The tested formulations were coded by a letter (mouthwashes A, B, and C, and oral gels D and E), thus blinding the experimenters about with regard to their compositions or the effect of the formulations. A negative control group included sterile phosphate-buffered saline solution (PBS).

### 4.2. Reconstituted Human Oral Epithelium (RHOE)

A total of 54 specimens of RHOE were used for the study. Specimens were shipped in 24-well plates containing agarose-nutrient transport medium. Upon arrival in the laboratory, the bag containing the RHOE specimens was opened under a sterile airflow hood. The specimens were extracted from the transport plate, and the agarose was removed. Then specimens were then placed in 6-well plates with nutrient medium (RPMI 1640 medium, supplemented with 20.0% fetal bovine serum, 1.0% L-glutamine, and 1.0% penicillin/streptomycin). Before testing, the culture plates were incubated overnight at 37 °C, in a 5% CO_2_ atmosphere and saturated humidity.

### 4.3. Bioreactor

A Modified Drip-flow Bioreactor (MDFR) was used for this study. The device is a modification of a commercially available Drip Flow Reactor (DFR 110; BioSurface Technologies, Bozeman, MT, USA). The modified design allowed the placement of customized trays on the bottom of the flow cells and the immersion of RHOE specimens into the surrounding flowing medium (Figure 1). Such changes allowed the use of nutrient medium at a continuous flow rate. All tubing and specimen-containing trays of MDFR were sterilized before the experiment using a chemiclave with hydrogen peroxide-based sterilization system (Sterrad; ASP, Irvine, CA, USA). By limiting the maximum temperature to 45 °C, heat-related damage to the whole system is avoided. The MDFR was then assembled inside a sterile hood. The specimens were cut out from their carrier using a sterile scalpel and tweezers and placed into eight polytetrafluoroethylene (PTFE) trays containing four holes, which fixed them and exposed their surfaces to the flow medium. All trays were fixed on the bottom of each of the flow chambers of two MDFRs running in parallel and immediately inoculated with a fresh nutrient medium. The MDFR was transferred into an incubator operating at 37 °C, 5% of CO_2_, and 100% relative humidity atmosphere. A multichannel, computer-controlled peristaltic pump (RP-1; Rainin, Emeryville, CA, USA) was turned on and used to provide a constant flow of nutrient medium through the flow cells. The flow rate was set to 9.6 mL/h.

### 4.4. Test Procedures

After an additional 24 h, the pump was stopped, and specimens were randomly allocated to one of three treatment groups (*n* = 18/group, three for each tested formulation and PBS reference, Figure 2). The first treatment included the exposure of specimens to the corresponding formulation for 30 min to evaluate possible cytotoxic effects. The second treatment included a preliminary treatment of the specimens with 1 wt% H_2_O_2_ for 90 s to simulate intense oxidative stress. Specimens were then rinsed for 5 min with sterile PBS and exposed to the corresponding formulation for 30 min to evaluate possible protective/repairing effects of the formulations against oxidative stress. The third treatment included the exposure of specimens to the corresponding formulation for 30 min, then rinsing with sterile PBS for 5 min, and subsequent exposure of the specimens to the 1 wt% H_2_O_2_ for 90 s. This latter treatment evaluated a possible preventive effect of the tested formulations in reducing oxidative stress damages. After each treatment, all RHOE specimens were extensively rinsed with sterile PBS, then immediately cut into four equal parts using a sterile scalpel and tweezers and processed for MTT assay, confocal microscopy (CLSM), and light and transmission electron microscopy analysis.

### 4.5. Specimen Evaluation

After each treatment, RHOE specimens from each mouthwash and gel formulations (Table 1) and the reference (PBS) were extensively rinsed with sterile PBS, then immediately cut into four equal parts using sterile scalpels and tweezers (Figure 2). They were quantitatively evaluated using MTT viability assay (*n* = 6, two for each specimen). Morphological analysis of the specimens was performed using Confocal Laser-Scanning Microscopy (CLSM) imaging (*n* = 3) and histological evaluation (*n* = 3) using light microscopy and Transmission Electron Microscopy (TEM) imaging.

### 4.6. MTT Assay

Cell viability was evaluated via MTT viability assay [17]. The assay was performed as follows: two starter stock solutions were prepared by dissolving 5.0 mg/mL 3-(4,5)-dimethylthiazol-2-yl-2,5- difenyltetrazolium bromide (MTT) in sterile PBS, and 0.3 mg/mL of N-methylfenazinium methyl sulfate (PMS) in sterile PBS. The solutions were stored at 2 °C in lightproof vials until the day of the experiment when a fresh measurement solution (MS) was made by mixing on a 1:1:8 ratio, respectively, MTT stock solution, PMS stock solution, and sterile PBS. A lysing solution (LS) was prepared by dissolving 10 % *v/v* of sodium dodecyl sulfate and 50% *v/v* of dimethylformamide in distilled water. RHOE specimens subjected to MTT assay were placed inside the wells of a sterile, flat-bottomed 24-well plate. After that, 1 mL of MS was pipetted into each well, and the plates were incubated at 37 °C in lightproof conditions for 1 h. During incubation, electron transport across the cell membrane and, to a lesser extent, cellular redox systems converted the yellow MTT salt to insoluble purple formazan. The conversion was facilitated by the intermediate electron acceptor (PMS). The unreacted MS was then gently removed from the wells by aspiration. The formazan crystals were dissolved by adding 1 mL of LS into each well, followed by additional incubation under agitation at room temperature in lightproof conditions for 1 h. A total of 100 µl of the suspension was then removed from each well, and optical density (550 nm) was measured with a spectrophotometer (Genesys 10-S, Thermo Spectronic, Rochester, NY, USA).

### 4.7. CLSM Observations

For CLSM imaging, the RHOE specimens were stained using Syto 9 and DAPI dual staining (Invitrogen Ltd., Paisley, UK). The fluorescence from stained cells was observed using a CLSM (Eclipse Ti2 inverted CLSM, Nikon, Tokyo, Japan). Four randomly selected image stack sections were recorded for each RHOE specimen. Confocal images were obtained using a dry Plan Apochromat 20× (NA 0.75) objective and digitalized using Nikon proprietary software (NIS), at a resolution of 1024 × 1024 pixels and 1.0 zoom factor. For each image stack section, 3D-rendering reconstructions were obtained using Drishti 3D software.

### 4.8. Histological Evaluation

RHOE specimens undergoing histological analysis were fixed overnight in freshly prepared Karnovsky solution (2% paraformaldehyde, 2% glutaraldehyde in 0.1 M sodium cacodylate buffer). After rinsing in the cacodylate buffer, specimens were postfixed in 2% OsO_4_ and stained en block with 2% aqueous uranyl acetate. They were then dehydrated in a graded acetone series and embedded in Epon-Araldite resin (EMS, Hatfield, PA, USA). Semi-thin sections (0.5 µm) of each specimen were obtained by an ultramicrotome (Leica Supernova, Reichert Ultracut, Leica Microsystems, Wetzlar, Germany) and stained with toluidine blue [17]. Then, they were examined using a digitalized light microscope with a 63× Apochromatic objective NA 1.40, at a final magnification 630×, (Zeiss Axiophot microscope, Oberkochen, Germany) and by TEM (Zeiss EM10 microscope, Oberkochen, Germany), magnification 1600×.

### 4.9. Statistical Analysis

MTT assay dataset was preliminarily checked for normality of distribution (Shapiro–Wilk’s test) and homoscedasticity (Levène’s test). ANOVA followed by Tukey’s post-hoc test were used for each treatment to assess significant differences between groups (*p* < 0.05).

## 5. Conclusions

A mouthwash formulation with hydrolyzed DNA-RNA effectively protected against oxidative stress without additional enhancement by other bioactive molecules. Active compounds such as hyaluronic acid, β-Glucan, allantoin, bisabolol, ruscogenin, and essential oils showed a protective effect against oxidative stress, which was not synergistic with the one of DNA-RNA. Reduced, yet significant, cytotoxic activity of the tested formulations may have been caused by the incorporation of surfactant agents.

## Figures and Tables

**Figure 1 molecules-26-02976-f001:**
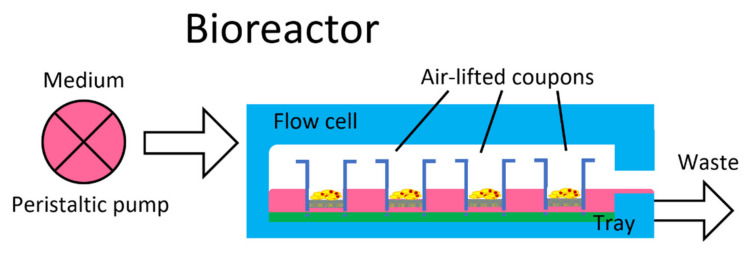
Functioning diagram of the coupon-bearing bioreactor. The peristaltic pump provides a constant flow of supplemented medium to the flow cells for the specified amount of time. The whole system is provided with a 5% CO_2_ supplemented atmosphere.

**Figure 2 molecules-26-02976-f002:**
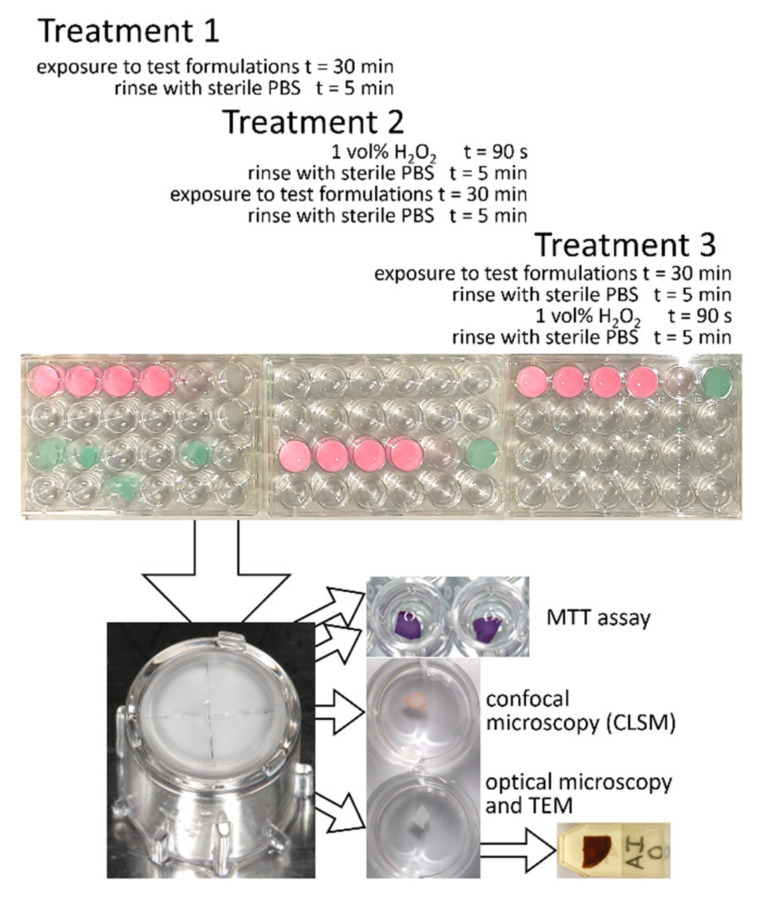
Diagram representing the treatment procedures and specimen evaluation. After each treatment, RHOE coupons were cut into four equal parts using a sterile scalpel blade and tweezers and subjected to quantitative assessment of the viable biomass (MTT assay), morphological and viability assessment (CLSM), and histological structural and ultrastructural analysis using light microscopy and TEM.

**Figure 3 molecules-26-02976-f003:**
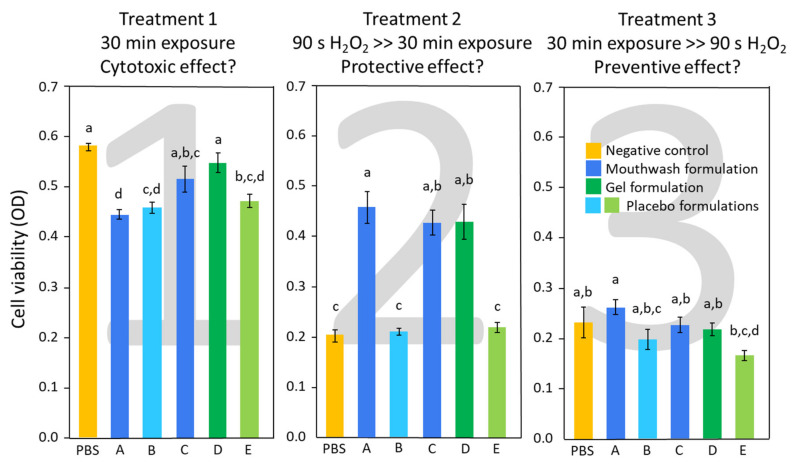
Viable and metabolically active RHOE cells. Results are displayed in optical density units as means ± 1 standard error (whiskers). Different superscript letters indicate statistically significant differences between groups (Tukey’s test, *p* < 0.05).

**Figure 4 molecules-26-02976-f004:**
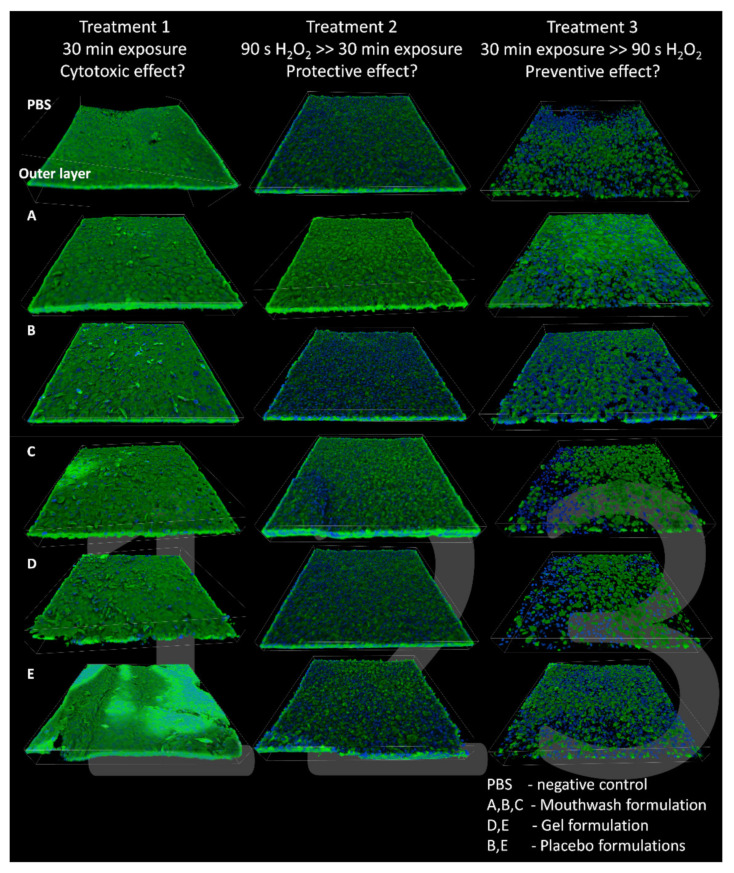
(**A**–**E**) 3D reconstructions of CLSM images (1270 × 1270 μm) of RHOE specimens. Live cells fluoresce bright green, whereas dead cells with compromised membranes show their nuclei in blue fluorescence. Each picture letter indicates the corresponding formulation, while each column represents one of the three treatments performed. Due to limited dye penetration, the external uppermost part of the epithelia could be visualized.

**Figure 5 molecules-26-02976-f005:**
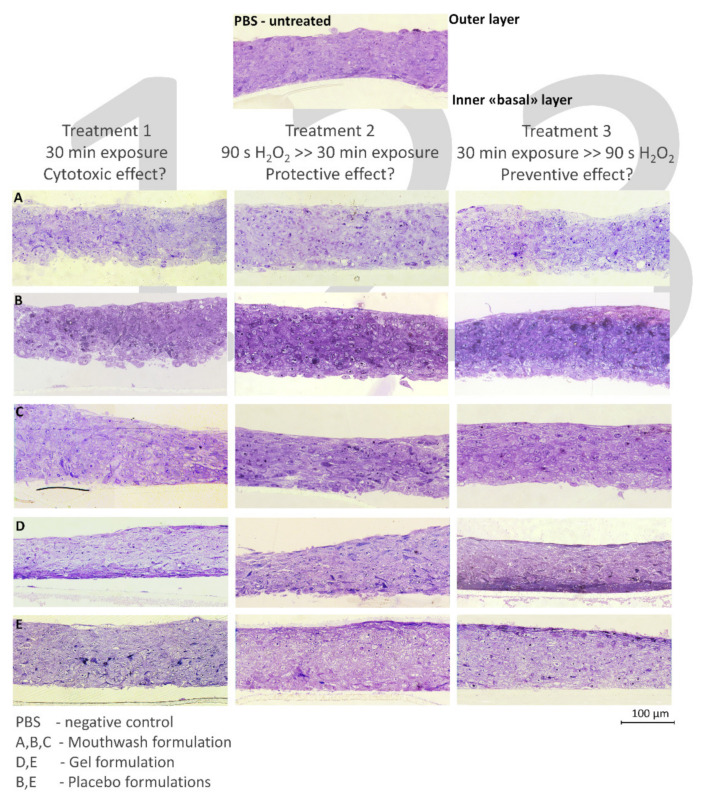
(**A**–**E**) Semi-thin sections of RHOE specimens. The outer layer of the epithelium is oriented upwards (Scale bar: 100 µm). Each picture letter indicates treatment with the corresponding formulation, while each column represents one of the three treatments performed. Higher magnification fields were acquired, then photo-stitching and contrast optimization were performed to obtain high-resolution fields spanning the whole thickness of each epithelium.

**Figure 6 molecules-26-02976-f006:**
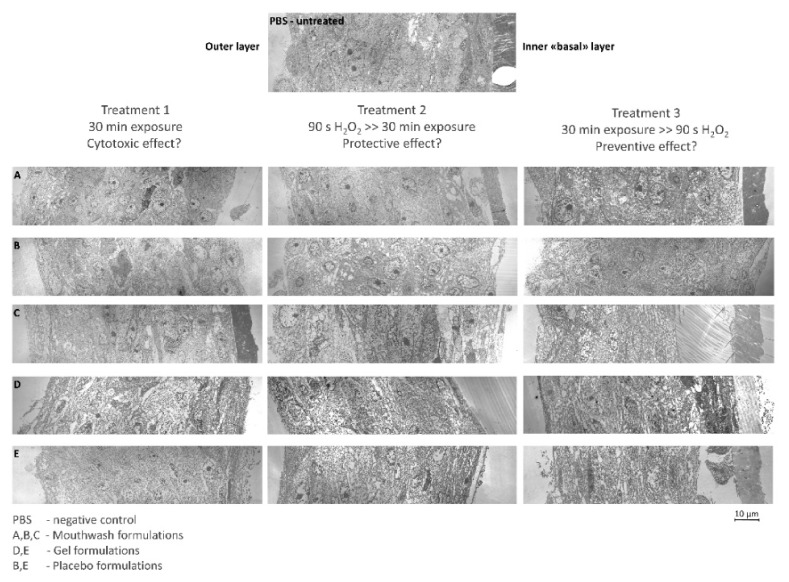
(**A**–**E**) TEM images of RHOE sections. The outer layer of the epithelium is oriented as shown in the upper image of PBS untreated. The different layers that characterize the human oral epithelium from the basal (**right**) to the keratinized surface layer (**left**) can be observed (Scale bar: 10 µm). Each picture letter indicates the corresponding formulation. Higher magnification fields were acquired, then photo-stitching and contrast optimization were performed to obtain high-resolution fields spanning the whole thickness of the epithelium.

**Table 1 molecules-26-02976-t001:** Ingredients of the formulations tested in the present study.

A (DNA-RNA Mouthwash Solution)	B (Placebo Mouthwash Solution)	C (Market-Ready DNA-RNA Mouthwash Solution)	D (Market-Ready DNA-RNA Gel Formulation)	E (Placebo Gel Formulation)
AQUA	AQUA	AQUA	AQUA	AQUA
DICAPRYLYL ETHER	DICAPRYLYL ETHER	DICAPRYLYL ETHER	PROPYLENE GLYCOL	PROPYLENE GLYCOL
COCO-CAPRYLATE/ CAPRATE	COCO-CAPRYLATE/ CAPRATE	COCO-CAPRYLATE/ CAPRATE	VP/VA COPOLYMER	
CETEARETH-20	CETEARETH-20	CETEARETH-20	CARBOMER	CARBOMER
CETYL PALMITATE	CETYL PALMITATE	CETYL PALMITATE	CELLULOSE GUM	CELLULOSE GUM
CETEARYL ALCOHOL	CETEARYL ALCOHOL	CETEARYL ALCOHOL	CALCIUM/SODIUM PVM/MA COPOLYMER	
CETEARETH-12	CETEARETH-12	CETEARETH-12		
GLYCERYL STEARATE	GLYCERYL STEARATE	GLYCERYL STEARATE		
XYLITOL	XYLITOL	XYLITOL		
PROPYLENE GLYCOL		PROPYLENE GLYCOL		
HYDROLYZED RNA		HYDROLYZED RNA	HYDROLYZED RNA	
HYDROLYZED DNA		HYDROLYZED DNA	HYDROLYZED DNA	
		HYALURONIC ACID	HYALURONIC ACID	
		ALLANTOIN	ALLANTOIN	
		GLYCYRRHETINIC ACID	GLYCYRRHETINIC ACID	
		BETA-GLUCAN	1,2-HEXANEDIOL	
		GLYCERIN	GLYCERIN	
		1,2-HEXANEDIOL	CAPRYLYL GLYCOL	
		CAPRYLYL GLYCOL	BETA-GLUCAN	
		RUSCOGENIN	RUSCOGENIN	
		BISABOLOL	BISABOLOL	
		LEPTOSPERMUM SCOPARIUM BRANCH/LEAF OIL	LEPTOSPERMUM SCOPARIUM BRANCH/LEAF OIL	
		MELALEUCA ALTERNIFOLIA LEAF OIL	MELALEUCA ALTERNIFOLIA LEAF OIL	
		O-CYMEN-5-OL	O-CYMEN-5-OL	
PHENOXYETHANOL	PHENOXYETHANOL	PHENOXYETHANOL	PHENOXYETHANOL	PHENOXYETHANOL
SODIUM BENZOATE	SODIUM BENZOATE	SODIUM BENZOATE	SODIUM BENZOATE	SODIUM BENZOATE
SODIUM SACCHARIN	SODIUM SACCHARIN	SODIUM SACCHARIN	SODIUM SACCHARIN	SODIUM SACCHARIN
			AMMONIUM GLYCYRRHIZATE	
CITRIC ACID	CITRIC ACID	CITRIC ACID		
			PEG-40 HYDROGENATED CASTOR OIL	PEG-40 HYDROGENATED CASTOR OIL
AROMA	AROMA	AROMA	AROMA	AROMA
C.I.16255	C.I.16255	C.I.16255	C.I. 42090	C.I. 42090

## Data Availability

The data presented in this study are available within the article text and figures.

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
