# Peer review of "Activity of Experimental Mouthwashes and Gels Containing DNA-RNA and Bioactive Molecules against the Oxidative Stress of Oral Soft Tissues: The Importance of Formulations. A Bioreactor-Based Reconstituted Human Oral Epithelium Model"

_molecules, 2021, doi:10.3390/molecules26102976_

Round 1

Reviewer 1 Report

This study aimed to investigate the effects of different formulations that contain hydrolyzed DNA-RNA on oxidative stress in bioreactor-based reconstituted human oral epithelium. This is an interesting topic. However, this work remains some major issues. My comments are as following:

  1. The introduction and discussion are relatively verbose. The authors should concentrate on the negative effects of ROS-induced oxidative stress on oral soft tissues and the benefits of hydrolyzed DNA and/or RNA.
  2. Lack of consistence regarding the results of cytotoxic effects and protective effects among formulations in different assays. For instance, MTT assay suggested A, B, and E were cytotoxic, but histological evaluation indicated the cytotoxicity of D and E.

3. The authors did not examine the formulations that contain active compounds other than hydrolyzed DNA-RNA, or the formulations that contain only surfactant. Thus, it is insufficient to support the conclusion regarding the effects of active compounds and surfactants on oral epithelia. 

Author Response

This study aimed to investigate the effects of different formulations that contain hydrolyzed DNA-RNA on oxidative stress in bioreactor-based reconstituted human oral epithelium. This is an interesting topic. However, this work remains some major issues.

We wish to thank the Reviewer for carefully looking at our study. We acknowledge that many parts of the manuscripts were not clear, or did not stay focused on the topic; therefore, we revised the whole paper to answer the raised points.

Point 1. The introduction and discussion are relatively verbose. The authors should concentrate on the negative effects of ROS-induced oxidative stress on oral soft tissues and the benefits of hydrolyzed DNA and/or RNA.

We apologize for not being clear in our original manuscript. In agreement with the Reviewer’s suggestions, we have reduced the two mentioned sections and concentrated on the oxidative stress effects. The discussion was targeted at the effects of the tested formulations on oral soft tissues.

Point 2. Lack of consistence regarding the results of cytotoxic effects and protective effects among formulations in different assays. For instance, MTT assay suggested A, B, and E were cytotoxic, but histological evaluation indicated the cytotoxicity of D and E.

We apologize to the Reviewer as probably our descriptions were not clear. We updated the discussion section as follows. “Our histological data suggested that gel formulation D, containing the same bioactive molecules, did not express similar activity compared to the mouthwash in protecting against oxidative stress. At the same time, MTT results for gel formulation D did not show cytotoxic activity against RHOE. Thickeners and film-formers may have reduced the access and availability of nutrients and oxygen rather than allowing better contact and delivery of the bioactive molecules carried by the gels. This mechanism may explain morphological damages found on RHOE specimens assessed for cytotoxicity after both gel treatments. On the contrary, MTT results showed cytotoxic effects of A, B, and E formulations. In fact, MTT is a colorimetric test based on the reduction of the tetrazolium salt at the plasma membrane level and, to a lesser extent, by cellular redox systems. Therefore, one may speculate that its response may not be initially related to the ultrastructural changes highlighted using TEM analysis, especially after the relatively short exposure time to the different formulations. For the same reason, a relatively high MTT reduction indicating the existence of a protective action against oxidative stress by the test gel formulation was not confirmed by the other morphological observations (CLSM, histological evaluation, and TEM). The use of both biochemical and morphological analytical methods is therefore paramount for an in-depth understanding of the effects exerted by bioactive principles.”

Point 3. The authors did not examine the formulations that contain active compounds other than hydrolyzed DNA-RNA, or the formulations that contain only surfactant. Thus, it is insufficient to support the conclusion regarding the effects of active compounds and surfactants on oral epithelia.

We apologize for not being clear in our original manuscript and for the missing information. From the data that we obtained (comparing, for instance, formulation A with B), we can observe that the hydrolyzed DNA and RNA solutions effectively protected against oxidative stress but were not helpful in its prevention. As shown in Table 2, we performed a thorough literature search for the activity of all specified ingredients. No other ingredient in A or B except hydrolyzed DNA-RNA was documented to protect against oxidative stress. Moreover, the reduced cytotoxic activity of the tested formulations may have been caused by incorporating surfactant agents. Again, surfactants were the only ingredients that are documented to have a cytotoxic effect. We have now clarified this point in the discussion and conclusion sections to stress this point, as suggested.

Reviewer 2 Report

The present study investigates the protective effects of DNA-RNA compounds formulated mouth wash against oxidative stress.  The results are rather descriptive and only evaluate cell viability (documented by two methods -- MTT test and fluorescent staining). H&E and TEM are additional descriptive methods showing cell morphology and again documenting the protective effects of the compounds on cell viability.  Measurements of anti-oxidative response elements (HO, SOD and GSH) is needed to validate the study hypothesis. 

Author Response

The present study investigates the protective effects of DNA-RNA compounds formulated mouth wash against oxidative stress.  The results are rather descriptive and only evaluate cell viability (documented by two methods -- MTT test and fluorescent staining). H&E and TEM are additional descriptive methods showing cell morphology and again documenting the protective effects of the compounds on cell viability.  Measurements of anti-oxidative response elements (HO, SOD and GSH) is needed to validate the study hypothesis.

We thank the Reviewer for the piece of advice. The suggested analysis is very interesting since it would allow providing further insights on the physiological reactions of oral epithelia to oxidative stress when testing protective formulations. As it is an interesting issue to be addressed in future studies, we added this suggestion to the discussion section in the revised version of the manuscript.

Round 2

Reviewer 1 Report

No comments

Author Response

Thank you very much! Appreciated your time.